



# Kinetic characteristics investigation of the Yingxingping rockslide based on discrete element method combined with discrete fracture network

Bo Liu[1,2], Yufang Zhang[1,2], Xiewen Hu[3], Jian Li[1,2], Kun Yuan[1,2], Kun He[3], Jian Cui[1,2,*], Zhenhua Yin[1,2]

[1] Railway Engineering Research Institute, China Academy of Railway Science Corporation Limited, Beijing 100081, China
[2] State Key Laboratory for Track Systems of High-speed Railway, China Academy of Railway Science Corporation Limited, Beijing 100081, China
[3] Faculty of Geosciences and Environmental Engineering, Southwest Jiaotong University, Chengdu 611756, China

*Correspondence to*: Bo Liu (liubogeology@gmail.com)

**Abstract.** The development of rock fractures on the mountain ridge in meizoseismal area may lead to fatal rockslides. This study focused on a catastrophic post-earthquake rockslide in Wenchuan, Southwestern China, to illustrate this geological phenomenon. On-site inquiries and aerial photography were used to ascertained the basic characteristics and determine three regions: the source area, transitional area, and depositional area. A three-dimensional discrete element method (DEM) with the discrete fracture network (DFN) was employed to assess the dynamic process of the rockslide. The sliding mass disintegrated quickly into smaller blocks, with those in the front-edge reaching the bottom of the slope earlier and experiencing higher acceleration. The maximum velocity and displacement of the sliding blocks were found to be 56.75 m/s and 508.61 m, respectively, lasting for approximately 104.45 seconds. The effects of fractures density and friction angle on kinetic characteristics were analysed, and a check dam was utilized to intercept the rockslide deposit in the debris flow gully. This study provides valuable information for assessing the kinetic process and preventing post-earthquake rockslides in meizoseismal areas.

## 1 Introduction

The northwest of the Sichuan Basin is recognised as one of the regions with high frequency of geological disasters in the world (Wu et al. 2012). Following a strong earthquake, the number and scale of geological disasters in meizoseismal areas typically exhibit an increase (Huang et al. 2012). As the result of an earthquake, cracks are formed in the rock mass within mountain ridges, which are often difficult to identify due to vegetation restoration, making the disasters that are triggered by them all the more devastating (Yan et al. 2020). Crack development can be classified into two types- retrogressive development, resulting in small-scale landslides that can be predictably prevented, and integral sliding that creates large-scale landslides which are difficult to prevent due to their suddenness (Fan et al. 2019). For instance, the Yigong rock avalanche in 2000 caused the damming of the Yigong Zangpo River. Although it was initially contained, the dam eventually failed after 62 days of outburst,



leading to a major outburst flood downstream in the Yarlung Zangpo rivers (Delaney et al. 2015). Another example is the 2017 Xinmo landslide that occurred in the intense tectonic area causing serious casualties and the loss of properties (Fan et al. 2017). The occurrence of numerous cracks on the rear edge of the mountain before the landslide highlights the need for improved proactive measures to be established to prevent such disasters.

The current research in the field of rockslides worldwide mainly focus on the initiation mechanisms and dynamic disaster-causing pattern of rockslides. Generally, field investigations and remote sensing interpretations are used to directly obtain the basic characteristics and some dynamic indicators of landslides. In order to reveal the dynamic features of rockslides and provide reference for engineering design and prevention, further quantitative analysis is necessary. For example, the Fahrbschung criterion and the Scheidgger model can be used to quickly obtain the motion characteristic parameters of

landslides (Evans et al. 2001). Models based on the principle of energy conservation can predict the sliding distance of debris flows (Okura et al. 2001). Laboratory physical simulation experiments can be used to study the deformation and failure mechanisms of landslides under different loads or conditions (Mangeney et al. 2010). However, due to the inability to proportionally scale down the gravity acceleration, it is difficult for these models to accurately match the forces experienced by real landslides in nature.

With the development of computer technology, numerical simulation has become an important tool in landslide research, especially its three-dimensional visualization simulation capabilities, as well as its outstanding computing power in analyzing failure mechanisms, stability analysis, and dynamic process inversion. Such as, the finite difference method has been employed for landslide simulation in several studies (Yin et al. 2015; Longoni et al. 2014). For example, to analyze the unstable slope in Mabian, Wei et al. (2019) employed PFC3D and obtained consistent results; however, the failure mechanism of rock mass

remained unexplored. In contrast, while two-dimensional discrete element method (DEM), DDA2D, used by Huang et al. (2019) to simulate the Xinmo landslide, easily revealed dynamic characteristics, certain details such as movement pathways were lost. There are often a large number of structural planes in the sliding mass of a rock landslide. These structural planes vary in location, size and occurrence, and their spatial distribution tends to be random network characteristics, which will control the landslide instability mode (Muaka et al. 2017). Hence, three-dimensional DEM has been proposed as a more

realistic method for simulation, allowing for the exploration of failure mechanics, pathway, and dynamic characteristics of sliding mass (Corkum and Martin 2004; Espada et al. 2018; Fan et al. 2019; Verma et al. 2019). Widely used for dynamic characteristics studies of rockslide, 3DEC, as a representative three-dimensional DEM, has been used to demonstrate the applicability and effectiveness of DEM (Liu et al. 2021; Wu et al. 2017). Liu et al. (2021), for instance, employed the discrete fracture network in 3DEC to investigate the effects of fracture density and internal friction angle on movement patterns.

Similarly, Wu et al. (2017) studied the erosion effect of Hsien-Du-Shan landslide on materials along the pathway. In previous studies, the conventional modelling approach for rock fractures often involved uniform subdivision of the sliding mass into cubes. Although this method can provide relatively accurate estimates of the accumulation range, it neglects the influence of the actual rock structure on the initiation and instability process of the landslide (Wu et al. 2017). Discrete Fracture Network (DFN) models are primarily based on the establishment of joint and fracture networks derived from field surveys and can also



use random vector distribution functions to generate other joints that cannot be fully measured (Havaej et al. 2016). Therefore, combining the discrete fracture network model with the discrete element method enables a more accurate simulation of the effects of the rock mass structure on the mechanisms and dynamic processes of slope initiation and instability.

In consideration of the aforementioned research, a DEM approach was employed to investigate the kinetic process of a post-earthquake rockslide. To exemplify this approach, the Yinxingping (YXP) rockslide in Wenchuan was intensively studied through field investigations, which allowed for the identification of the failure mechanism. The establishment of a real three-dimensional model of the YXP rockslide based on aerial photography enabled the use of discrete element numerical simulation, taking into account the high degree of weathering and fragmentation of the rock mass in the source area and the establishment of the fractures using discrete fracture network (DFN). The simulation results provided insight into the movement mode and dynamic characteristics of the post-earthquake rockslide in terms of velocity and displacement. Additionally, the influence of friction angle and density of the fractures on the kinetic process was discussed. Furthermore, a check dam was implemented to prevent the movement of the depositional materials in the YXP gully, and the DEM method was used to confirm the feasibility of this measure. This study contributes to a better understanding of the disaster mechanism and movement mode of post-earthquake rockslides, and it proposes practical prevention measures that can serve as a reference for the prevention and control of similar rockslides in meizoseismal areas.

## 2. Geological setting

### 2.1 Geological setting and the rockslide event

The YXP landslide, located in Yinxing village of Wenchuan County, Sichuan Province, China (Fig. 1a). It is situated in the north-western region of the Sichuan Basin and is influenced by three major faults. The Wenchuan Ms 8.0 earthquake had its epicenter 25km away from the rockslide. YXP rockslide is situated approximately 800m away from the gully mouth of YXP debris flow gully (Fig. 2a), where the G213 national highway and Dujiangyan-Wenchuan expressway are positioned. The terrain in the study area is steep and deep-cutting with mainly Proterozoic granite, diorite, and some hard rocks. The lithology of the area is relatively broken. The average annual precipitation in the study area is 1253.1 mm with a maximum daily precipitation of 269.8 mm. The incident occurred on July 20, 2018, during a strong rainstorm, causing the YXP rockslide to accumulate in the nether YXP gully, subsequently leading to the material sources for the debris flow in YXP gully. This event resulted in traffic interruption of the downstream National Road G213 and the blockage of the river, causing the loss of lives and property of the downstream residents.



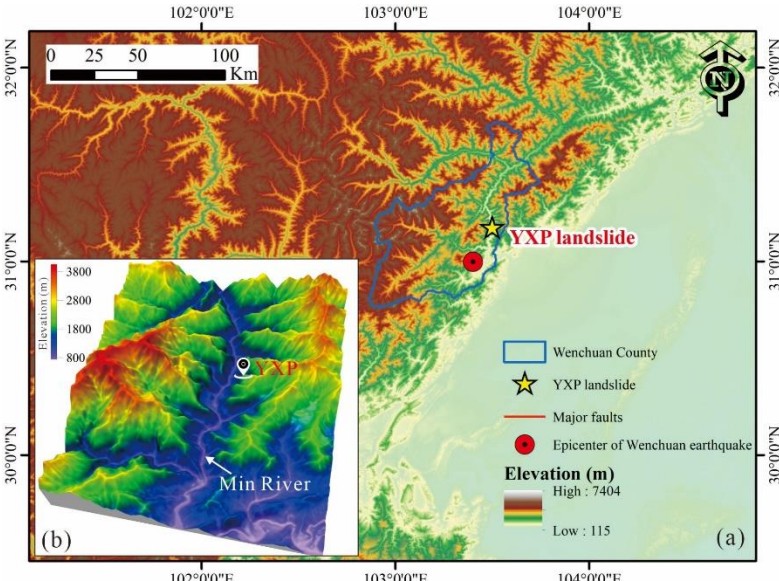

**Figure 1: The location (a) and landform (b) of the YXP rockslide.**

The YXP rockslide can be subdivided into three areas: the source area characterized by a steep slope of approximately 60°, the transitional area with a slope angle of 47° and a length of around 120 m, and the depositional area where the sliding mass converges (Fig. 2b, 3). The sliding region had a total area of 31419 m² and an average thickness of 5.8 m, resulting in a total volume of approximately $18.22 \times 10^4$ m³ (Fig. 3, 4a). During the Wenchuan Ms 8.0 earthquake, the source area experienced a significant number of crack formations (Fig. 2c). Subsequently, under rainfall, the rainwater infiltrated the fractures and increased the pore water pressure, eventually leading to instability and sliding of the mass (He et al. 2020). The deposition of rockslide materials in the gully of YXP debris flow inclined downstream due to the configuration of the bottom of the depositional area. Fortunately, most of the materials were entrapped by a check dam built in the gully beforehand (Fig. 4c).



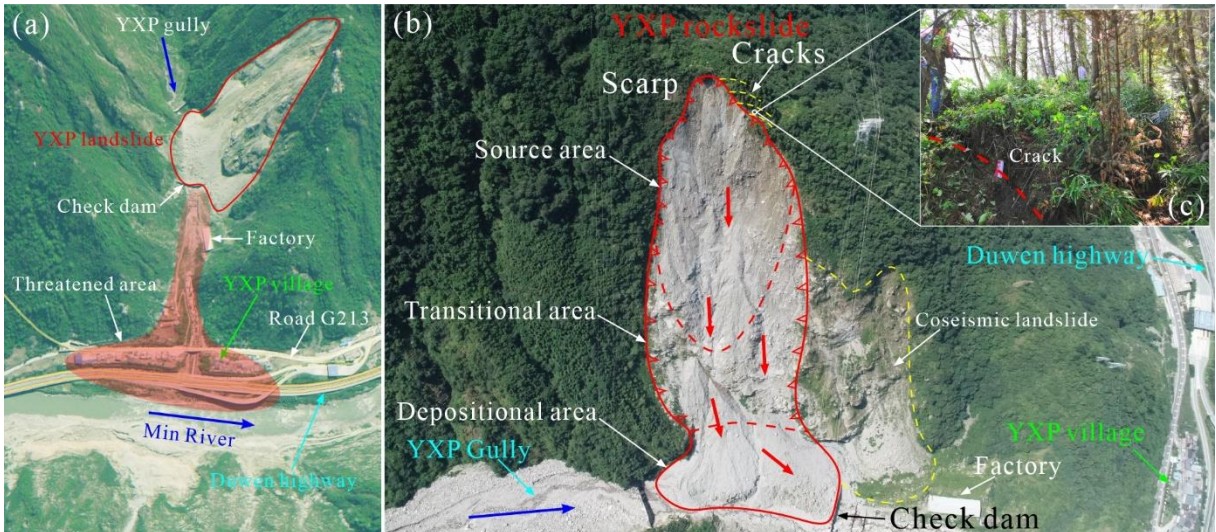

**Figure 2: (a) The threatened area of YXP rockslide: © Google Earth; (b) The top view of YXP rockslide taken by UAV.**

## 2.2 Failure mechanism

Affected by the Wenchuan Ms 8.0 earthquake, the rock mass in the landslide area was seriously broken, and the two sets of controlled fractures provided the material source for the rockslide (Fig. 4d). In addition, the hard and soft rocks in the study area were affected by weathering, erosion and other factors, which also create conditions for the occurrence of geological disasters such as collapse, landslide and debris flow (Huang et al. 2014). The climate in landslide area was subtropical monsoon climate, the distribution of annual precipitation was uneven, which was mainly concentrated in summer. The accumulated

rainfall in July, 2018 reached the peak of 203.3 mm. As the slope was soften by continuous heavy rainfall, the shear strength of the rock mass decreased, the slope was more prone to slip (Liu et al. 2021). Based on the findings from the field investigation, the sliding plane located at the rear edge of the YXP landslide was observed to be clear and smooth (Fig. 4a). Consequently, the deposit from the sliding considerably obstructed the gully situated at the foot of the slope (Fig. 4c). An in-depth examination of the landslide revealed that it was mainly comprised of quartzite gravel, which had minimal impact on the small-discharge

stream in the YXP gully due to its large particle size and large aperture gaps. Nonetheless, if a massive debris flow ensues, the discharge will increase significantly, resulting in the obstruction effect, ultimately leading to severe damage to buildings situated downstream (Hu et al. 2017).





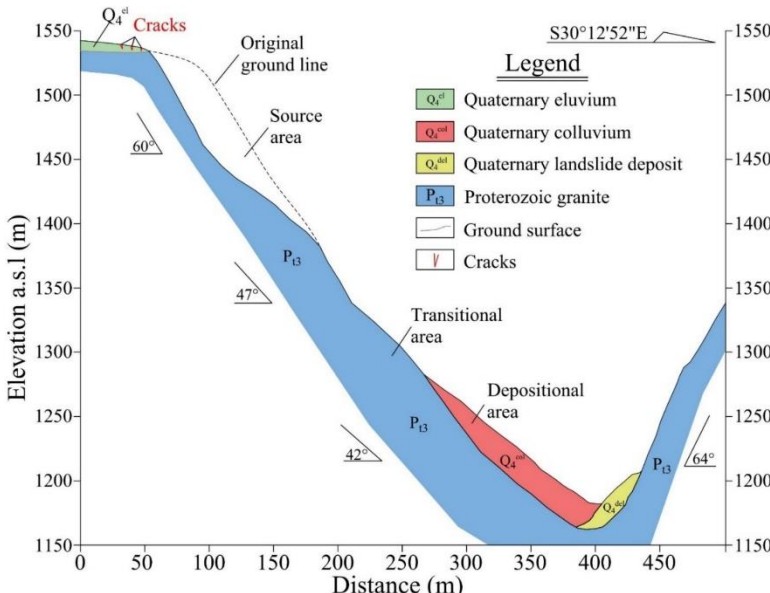

**Figure 3: The main longisection of the YXP rockslide.**

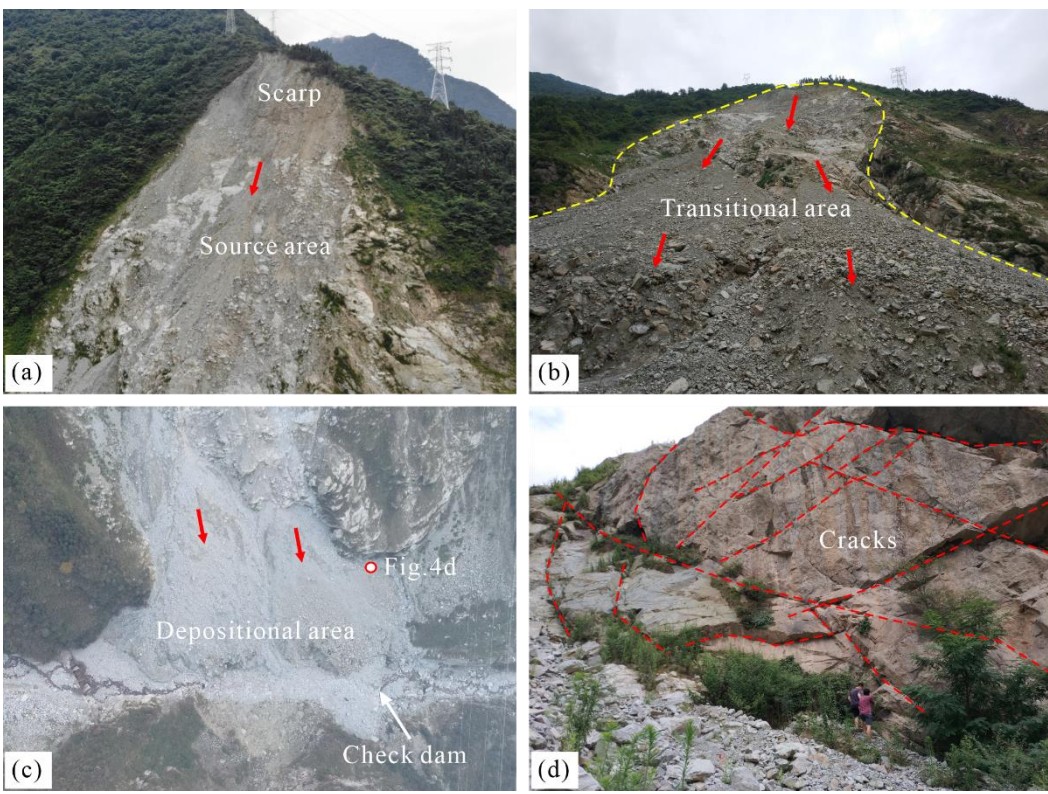

**Figure 4: A series of photographs include the scarp (a), transitional area (b), depositional area (c), and fractures (d).**




## 3 Method and modelling

### 3.1 The discrete element method

The three-dimensional DEM has a rapid development with the researchers' efforts around the world (Kim et al. 2007; Lemos
et al. 2015; Ma et al. 2021). It is appropriate for simulating highly nonlinear behavior because an explicit solution was utilized
(Wang et al. 2006). The built-in discrete fracture networks are employed to simulate fractures in rock mass. Each block could
be discretized into many tetrahedral elements (Einstein et al. 1983). The contacts in 3DEC were modelled scientifically. The
fracture normal stiffness, $K_n$, is parallel with the normal contact damper in the normal direction. In addition, the fracture
tangential stiffness, $K_s$, is parallel with the tangential contact damping connected with Mohr Coulomb slider, $S_{m-c}$, in the
tangential direction (Fig.5).

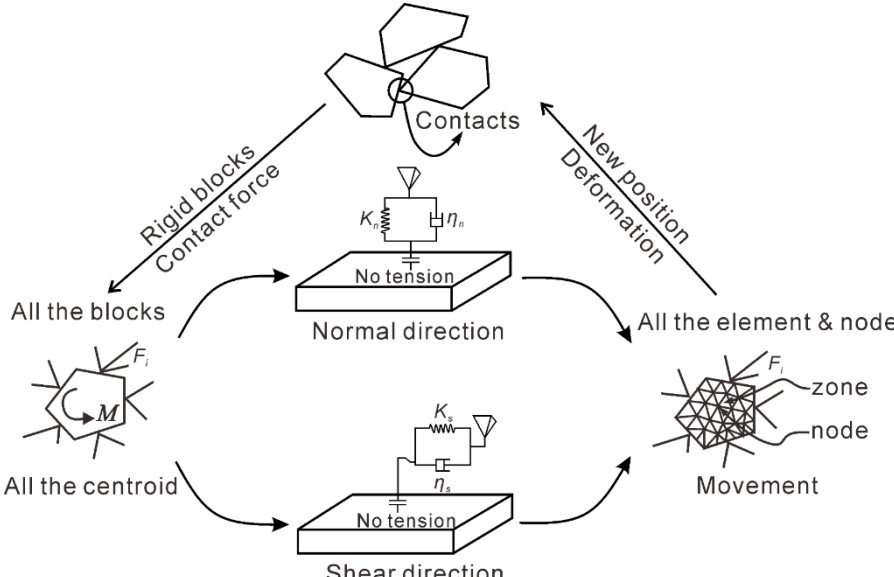

**Figure 5 The calculation circle in DEM.**

A fundamental equation of motion in 3DEC has been formulated, as stated by Liu et al. (2020):

$$m\ddot{u}(t) + c\dot{u}(t) + ku(t) = f(t) \tag{1}$$

Where $m$ is the mass; $u$ is the displacement; $t$ is the timestep; $c$ is the viscous damping coefficient; $k$ is the stiffness coefficient;
$f$ is the external force of the element.

The relationships among normal force increment $\Delta F_n$, shear stress increment $\Delta F_s$ and displacement increment (normal
displacement increment $\Delta U_n$ and shear displacement increment $\Delta U_s$) are as follows:

$$\Delta F_n = K_n \Delta U_n A_c,$$

$$\Delta F_s = K_s \Delta U_s A_c. \tag{2}$$





Where $K_n$ and $K_s$ are the normal and tangential stiffness of joints [stress/displacement]. $A_c$ is contact area. Direct calculation of contact stress is possible by considering the contact force and area. The total normal force and shear vector are subsequently updated and adjusted based on the contact constitutive relation.

$$F_n := F_n + \Delta F_n,$$


$$F_s := F_s + \Delta F_s. \tag{3}$$

Where $F_n$: and $F_s$: are normal and shear force of the next step. 3DEC adopts Mohr-Coulomb friction law, the formula for calculating the maximum shear force is:

$$F_{max}^s = cA_c + F^n \tan\varphi \tag{4}$$

Where $c$ is cohesion [stress]; $\varphi$ is friction angle. Then the contact force is applied to the force and moment acting on the center

of two blocks.

The translational motion equation of a single block can be expressed as:

$$\ddot{x}_i + \alpha\dot{x}_i = F_i/m + g_i \tag{5}$$

Where $\ddot{x}_i$ is velocity of the center of mass; $\alpha$ is viscous damping constant (proportional to mass); $F_i$ is the sum of force acting on an object (from the contact of blocks and the applied external forces);m is mass of element;$g_i$ is gravity acceleration

vector。

The Euler equation was used for describing the rotational motion of undamped rigid elements. Because the dynamic analysis of rock mechanics system needs deformable blocks, the Euler equation of rotation can be simplified. Therefore, there is only one approximate moment of inertia $I$ and $L$ are calculated based on the average distance from the center of mass to the vertex of the block. By inserting the viscous damping term, the equation becomes as follows:


$$\dot{\omega}_i + \alpha\omega_i = \frac{M_i}{I} \tag{6}$$

Velocity $\omega_i$ and total moment $M_i$ are now applied to the global axis. The motion equation of the block is derived by using the central finite difference method. The following equations describe the values of translation and rotation velocities in the $t$-time.

$$\dot{x}_i^{(t)} = \frac{1}{2}\left[\dot{x}_i^{(t-\Delta t/2)} + \dot{x}_i^{(t+\Delta t/2)}\right],$$

$$\omega_i^{(t)} = \frac{1}{2}\left[\omega_i^{(t-\Delta t/2)} + \omega_i^{(t+\Delta t/2)}\right]. \tag{7}$$

Acceleration is calculated as:

$$\ddot{x}_i^{(t)} = \frac{1}{\Delta t}\left[\dot{x}_i^{(t-\Delta t/2)} + \dot{x}_i^{(t+\Delta t/2)}\right],$$




$$\dot{\omega}_i^{(t)} = \frac{1}{\Delta t}\left[\omega_i^{(t-\Delta t/2)} + \omega_i^{(t+\Delta t/2)}\right]. \tag{8}$$

These expressions are substituted into translation motion (Eq. 5) and rotation motion (Eq.6), respectively, and then the velocity of $[t + (\Delta t\,/2)]$ moment is solved:

$$\dot{x}_i^{(t+\Delta t/2)} = \left[D_1\dot{x}_i^{(t-\Delta t/2)} + \left[\frac{F_i^{(t)}}{m} + g_i\right]\Delta t\right]D_2,$$

$$\omega_i^{(t+\Delta t/2)} = \left[D_1\omega_i^{(t-\Delta t/2)} + \left[\frac{M_i^{(t)}}{m}\Delta t\right]\right]D_2. \tag{9}$$

Coefficient $D_1$ and $D_2$ are expressed:

$$D_1 = 1 - (\alpha\Delta t/2),$$

$$D_2 = 1/[1 + (\alpha\Delta t/2)]. \tag{10}$$

Increments of translation and rotation, $\Delta x_i$ and $\Delta\theta_i$ are expressed as:

$$\Delta x_i = \dot{x}_i[t + (\alpha\Delta t/2)]\Delta t,$$

$$\Delta\theta_i = \omega_i[t + (\alpha\Delta t/2)]\Delta t. \tag{11}$$

The position of the center of mass is updated to:

$$x_i^{(t+\Delta t)} = x_i^{(t)} + \Delta x_i \tag{12}$$

The new position of the vertex of the blocks is given by the following formula:

$$x_i^{\nu(t+\Delta t)} = x_i^{\nu(t)} + \Delta x_i + e_{ijk}\Delta\theta_j[x_k^{\nu(t)} - x_k^{(t)}] \tag{13}$$

When the block motion update is completed, the sum of all block forces and moments, $F_i$ and $M_i$ in each cycle will be reset to zero for the next calculation.

## 3.2 Model setting

We established a three-dimensional model of the rockslide through the UAV aerial photography and digital elevation model of ALOS satellite in Japan. The model was 1320 m long, 1065 m wide and 700 m high (Fig. 6a). According to the measurement, the longitudinal gradient of YXP gully in the landslide depositional area was about 14 degrees. In order to understand the kinetic characteristics of different regions of the sliding mass in the movement process, we set 10 monitoring points on the section A-A′ and B-B′, and monitored the velocity, total displacement and horizontal displacement at the same time (Fig. 6b, d).





According to the field investigation, we measured the dips and dip directions of 423 groups of rock mass fractures, and drew the corresponding Fisher distribution map. We found two sets of controlling fractures. The dip direction and dip angle of one set are 286° and 78°, and the dip direction and dip angle of another set are 62° and 32°, respectively (Fig. 6c). Then, we established a DFN model using built-in Fish language based on the collected data (Havaej et al. 2016), and the parameter

settings of fracture sets are shown in Table 1.

**Table 1: Summary of parameters used to generate the fractures of YXP rockslide. Where $k$ is the Fisher constant. $P_{32}$ is volumetric density of fractures. $P_{10}$ is linear density of fractures. $a$ is the scaling exponent. The length is in the range of $l_{min}$ to $l_{max}$.**

| Fractures | Property | Distribution | Parameters |
|---|---|---|---|
| Fracture set 1 | positions | uniform | positions generated in all of space |
| | orientations | Fisher | dip 78°, dip direction 286°, $k$=400 |
| | lengths | power law | scaling exponent: $a$=1.2, $l_{min}$ = 15, $l_{max}$ = 300 |
| | density | $P_{32}$ | 0.126 |
| Fracture set 2 | positions | uniform | positions generated in all of space |
| | orientations | Fisher | dip 32°, dip direction 62°, $k$=400 |
| | lengths | power law | scaling exponent: $a$=1.2, $l_{min}$ = 30, $l_{max}$ = 450 |
| | density | $P_{32}$ | 0.085 |
| Background fractures | positions | uniform | positions generated in all of space |
| | orientations | bootstrapped | $k$=200 |
| | lengths | power law | scaling exponent: $a$=1.0, $l_{min}$ = 10, $l_{max}$ = 500 |
| | density | $P_{10}$ | 0.240 |

**Figure 6: The three-dimensional model of YXP rockslide (a) and monitoring points in source area (b). Fisher distribution of in-site**
**measured fractures (c) and sliding mass cut by DFN (d).**




After that, the DFN model was used to cut the rock mass to form a fractured rock mass (Fig. 6d). The number of fractures and cut blocks were 3546 and 27777, respectively. The bedrock was fixed in the calculation process. Because the sliding mass and the bedrock were hard rocks, we set their constitutive models as elastic models. Then, according to the experience of similar areas, the value of parameters of sliding mass, bed rock and fractures were assigned before the simulation (Liu et al. 2021), and the values of parameters are shown in Table 2. Since the YXP rockslide was triggered by rainstorm, we assume that it was fully saturated when it initiates, so the parameters of the model were all saturation parameters.

**Table 2: The simulation parameters used in the kinetic characteristic analysis.**

| Parameters | | Value |
|---|---|---|
| Bed rock | Tensile modulus (GPa) | 3.50 |
| | Poisson's ratio | 0.35 |
| | Density (kg/m$^3$) | 2450 |
| Sliding mass | Tensile modulus (GPa) | 1.20 |
| | Poisson's ratio | 0.25 |
| | Density (kg/m$^3$) | 2550 |
| Fractures | Cohesion (kPa) | 0.00 |
| | Friction angle (degrees) | 8.00 |
| | Normal stiffness, $K_n$ (MPa/m) | 3.00 |
| | Shear stiffness, $K_s$ (MPa/m) | 3.00 |
| | Local damping ratio, $\alpha_d$ | 0.03 |
| Model | Number of fractures | 3546 |
| | Number of sliding mass blocks | 27777 |

## 3 Result

### 3.1 Characteristic of initiation and movement process

The displacement of landslides is a significant feature that can provide information about the position and movement of the sliding mass over time, as well as the accumulation characteristics of the sliding mass (Wang et al. 2019). The displacement characteristics of the YXP landslide are showed in Fig. 7. The middle part of the sliding mass start moving in a saturated condition, reaching the maximum displacement of 41.4m at 7.46 seconds (Fig. 7a). At 13.99 seconds, the sliding mass has completely disintegrated and move in a large scale, achieving the displacement of 207.87m (Fig. 7b). At 21.45 seconds, the rocks in the front edge of the sliding mass began to accumulate at the foot of the slope, which facilitated the movement of the middle and rear blocks into the deposition area with acceleration and collision resulting in a significant increase in deposition volume (Fig. 7c and 7d). The displacement of the rocks at $t$=39.17s reached 457.93m. The displacement distribution in the





depositional area indicated that the material at the front edge of the sliding mass reached the trench bottom and accumulated earlier, and the material at the rear edge accumulated upward in turn (Fig. 7e). When $t$ = 104.45s, most of the rocks had

completed their movement, while a few rocks remained in the source area. The left edge of the fan-shaped deposit inclined toward the downstream (Fig. 7f).

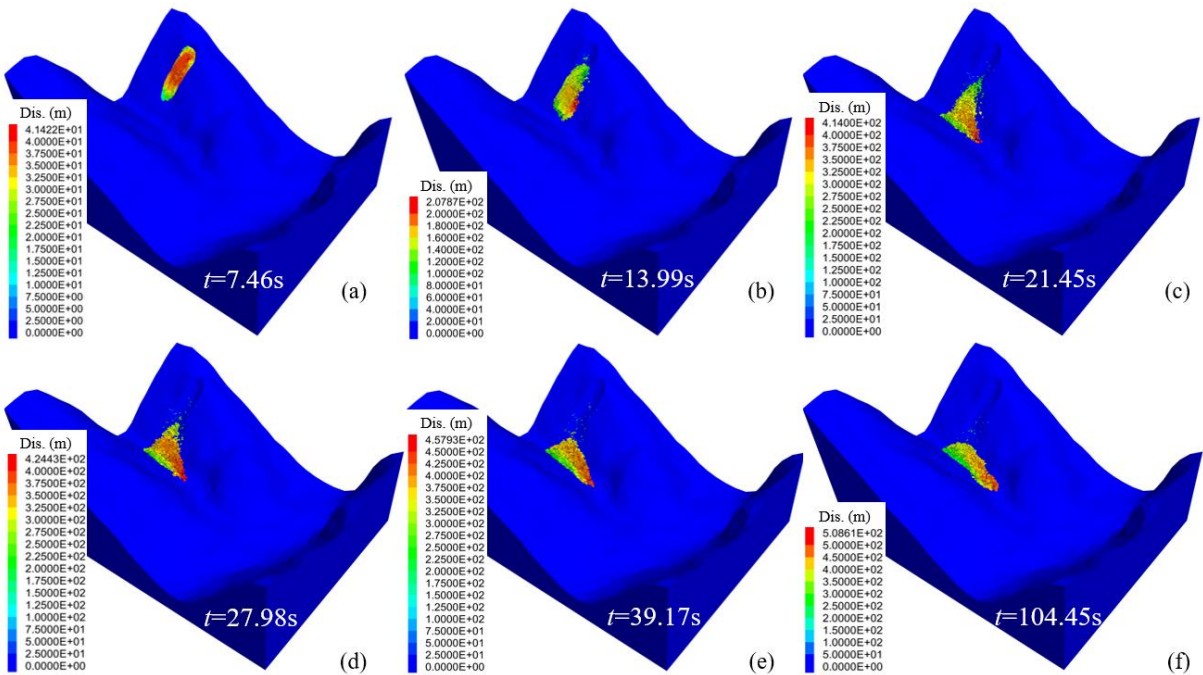

**Figure 7: The displacement figures of YXP rockslide when the time reached 7.46s (a), 13.99s (b), 21.45s (c), 27.98s (d), 39.17s (e), 104.45s (f).**

The displacement monitoring results of different points are shown in Fig. 8. The displacement of the monitoring points at the front edge and right side of the sliding mass had no obvious increase after reaching the bottom of the gully at about 20 s, indicating that was close to stop. However, the displacement of monitoring points 1, 2, 8 on rear edge and point 7 on left sides of the sliding mass still increased gradually after reached the bottom of the gully (Fig. 8b). As the monitoring point 7 was located at the left side of the sliding mass, it doesn't stop when reached the bottom of the gully lately because of the oblique

terrain. Influenced by the inherent slope of deposit and gully, the horizontal X-direction displacement of monitoring point 7 was larger than other points (Fig. 8b). These results indicated that the depositional blocks will keep moving to the downstream of the YXP gully because of the slope of the gully. If there were no prevention works, these materials will not stop until reaching a gentle region.



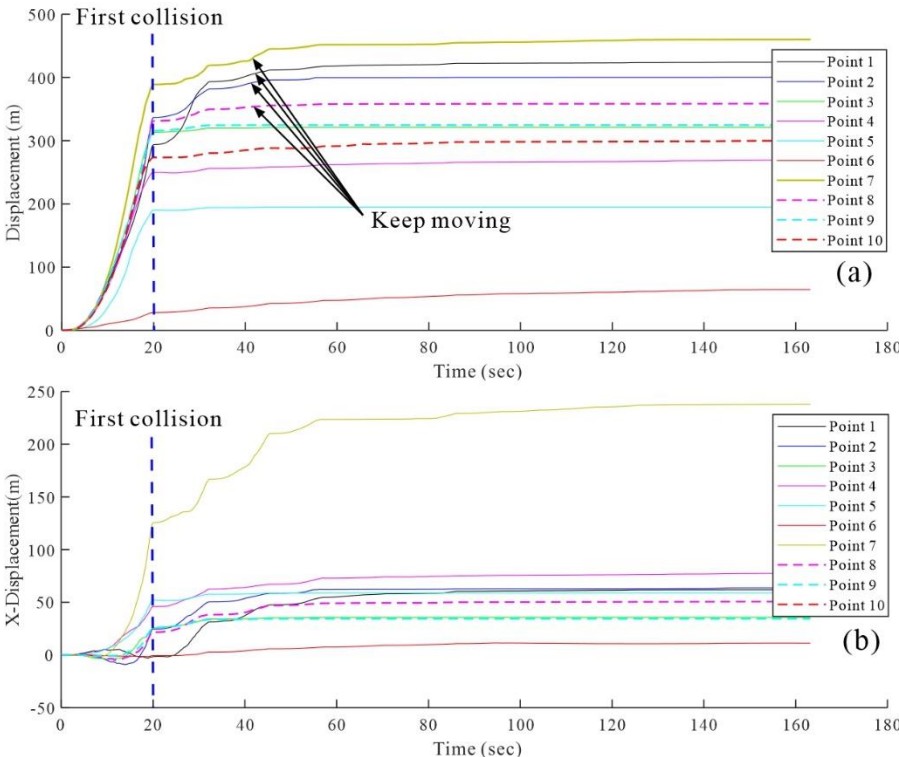

**Figure 8: Displacement (a) and x-displacement characteristics (b) of each monitor point.**

## 3.2 Characteristic of velocity

The velocity of rockslides is a critical parameter as it reflects the energy exchange of rock blocks during dynamic processes (Wang et al. 2020). The initial velocity after 6 seconds of the rockslide was measured at a maximum of 16.34 m/s for single surface blocks, with an average velocity of under 10 m/s (Fig.9a). At 13.99s, sliding rocks disintegrated to form a scarp at the rear edge (Fig. 9b). As time progressed to 21.45 seconds, the sliding mass showed downstream diffusion with increasing velocity. The maximum measured velocity reached 56.75 m/s, and the overall average velocity was close to 30 m/s. The sliding mass began to accumulate by the 27.98 second. Consequently, the velocity of material that previously reached lower regions of the gully rapidly decreased to zero, whereas the material at the rear edge continued to slide to the bottom (Fig. 9d). At 39.17 seconds, the deposition near the downstream began to expand, with occasional scattered falling blocks from the rear edge. The maximum velocity of these blocks was 8.94 m/s. By the 104.45nd second, the deposit near the downstream region exhibited a velocity of less than 0.25 m/s, indicating the sliding materials will stop here.

The monitoring results of profile 1-1′ showed that the maximum velocity of the monitoring points located in the middle and rear edge of the landslide was higher (such as point 3, 8 and 9), reaching 50.25 m/s, and the initiation time of these blocks was also earlier (Fig.10a). However, it should be noted that although the maximum velocity of monitoring points 1 and 2 was small, their velocity recovery ability after collision was strong. The reason was that it was located on the surface of the rear edge of





the sliding mass and kept above of the deposit during the movement, so its recovery ability was less limited. The monitoring results of section 2-2′ show that the kinetic energy loss of the blocks in the middle sliding mass was large after the collision because they arrived at the slope bottom and accumulate earlier (such as monitoring points 3 and 9) (Fig.10b). The energy recovery of the left and right blocks was faster after the collision (monitoring points 6 and 7), the reason might be that the

blocks on both sides arrived at the foot of the slope later and accumulated in the upper layer of the deposit. Therefore, the maximum velocity of the blocks initiated earlier was higher, while the velocity recovery ability of the blocks arriving at the foot of the slope earlier was weak due to the influence of the overburden (Fig.10b). The kinetic energy recovery of the blocks in rear edge was stronger because it stopped and accumulated on the surface of the deposit.

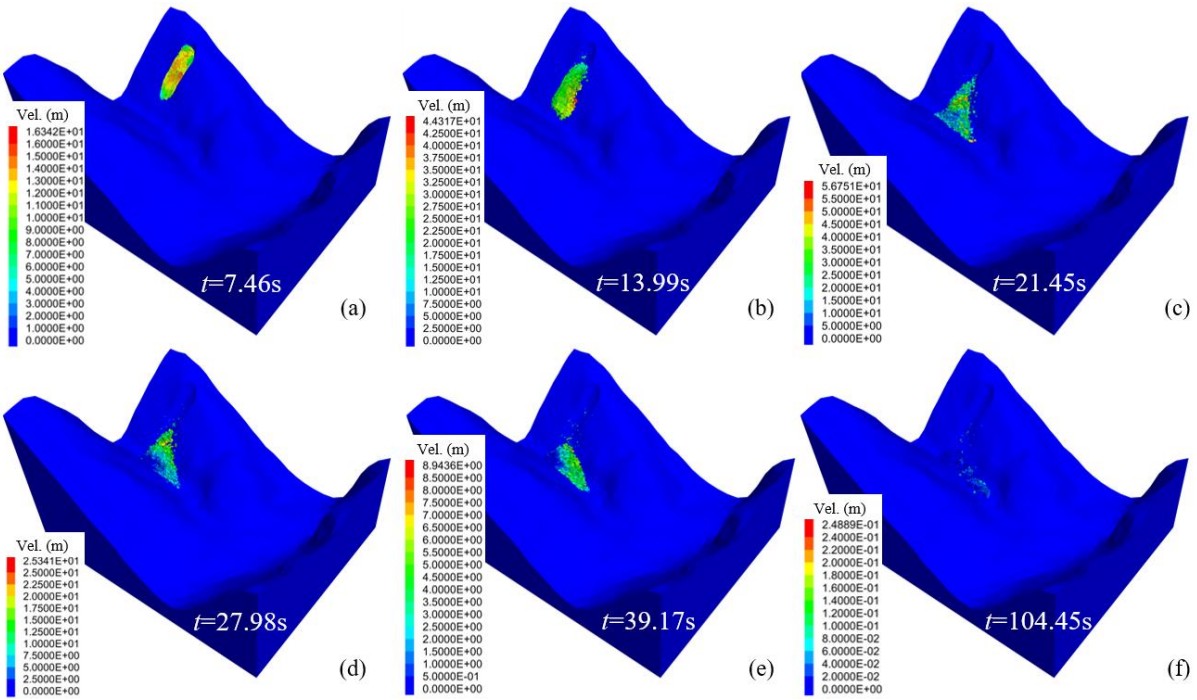

**Figure 9: The velocity figures of YXP rockslide when the time reached 7.46s (a), 13.99s (b), 21.45s (c), 27.98s (d), 39.17s (e), 104.45s (f).**



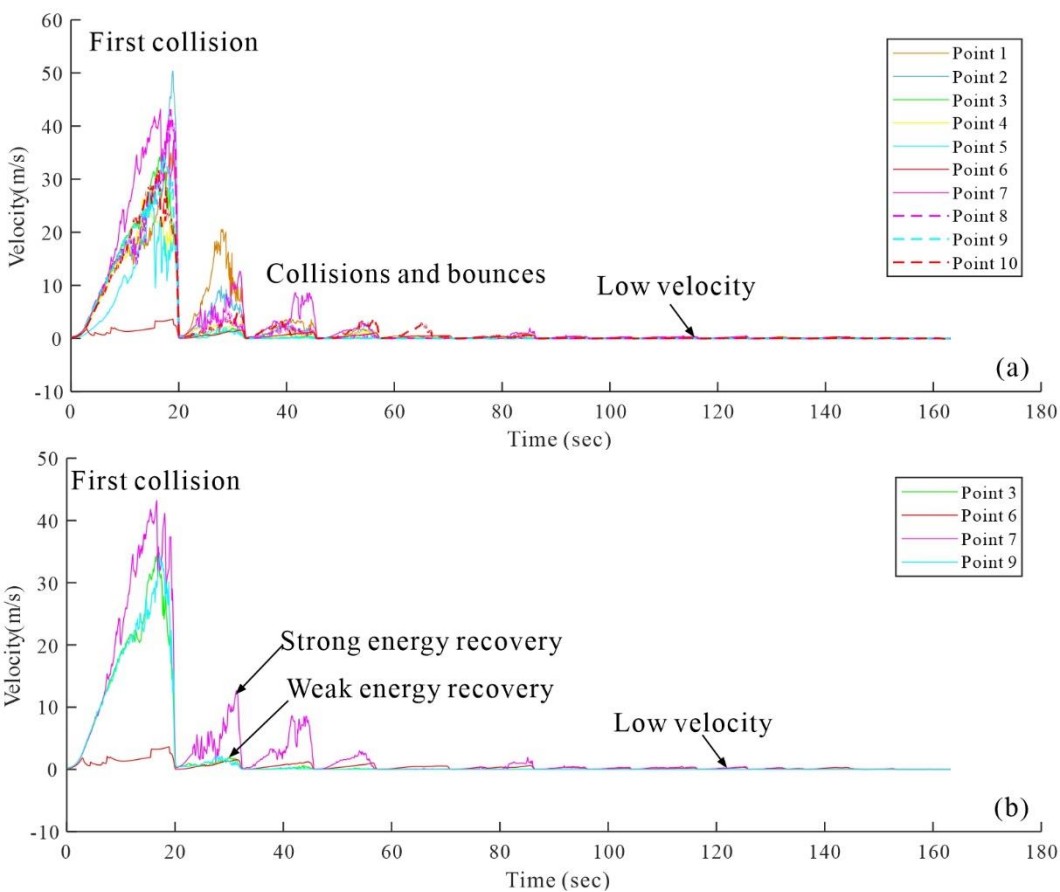

**Figure 10: Velocity curves of monitoring points of section a-a' (a) and section b-b' (b).**

## 4 Discussion

### 4.1 Effect of internal fraction angle of fractures

The kinetic energy loss of sliding mass is mainly affected by the collision and friction, so it is necessary to analyze the sensitivity of internal friction angle of fractures to simulation results (Lin et al. 2021). Keeping other conditions unchanged, we set the internal friction angle to 8°, 12° and 16°, respectively, and monitored the velocity of point 9. Fig. 11 exhibits the simulation results with different internal friction angles. The movement time of sliding mass increased with the increase of fractures internal friction angle, which were 58.64s, 75.42s and 104.75s, respectively. As the increase of the internal friction angle, the depositional area also increased. When $\varphi = 8°$, the maximum movement distance of the sliding mass reached about 753m due to the low internal friction angle and low slope of the depositional area (Fig.11a, d), reaching the residential area at the outlet of the YXP gully. Under rainfall conditions, due to the water-convergence effect, these loose solid materials will be transformed into debris flows, causing devastating damage to the villages. When $\varphi = 16°$, the velocity of blocks at the front





edge of the deposit decelerate to zero (Fig. 11f), and only a small number of blocks at the rear edge continued to move towards the deposit (Fig. 11c, f). When $\varphi$ =12°, the simulation results were consistent with the actual situation. the front blocks still had a velocity of less than 0.25m/s at 104.45 s (Fig.11b, e).

The monitoring results of blocks in middle sliding mass show that the maximum velocity increased with the decrease of internal friction angle, the bigger is, and it moves to the foot of the slope earlier. The maximum velocities of monitoring point 9 were
58.74 m/s, 42.35 m/s and 33.46 m/s, respectively under different internal friction angles. In addition, when the angle is 8°, the velocity recovery degree of the monitored block is the largest after the first collision. Then the rock mass continues to slide along the gully until it finally stops (Fig. 12). In the other two conditions, the energy of the monitored block is almost lost after the first collision, and only in condition 2, the block still has a weak velocity. In the other two cases, the energy recovery of the monitored block is weak after the first collision.

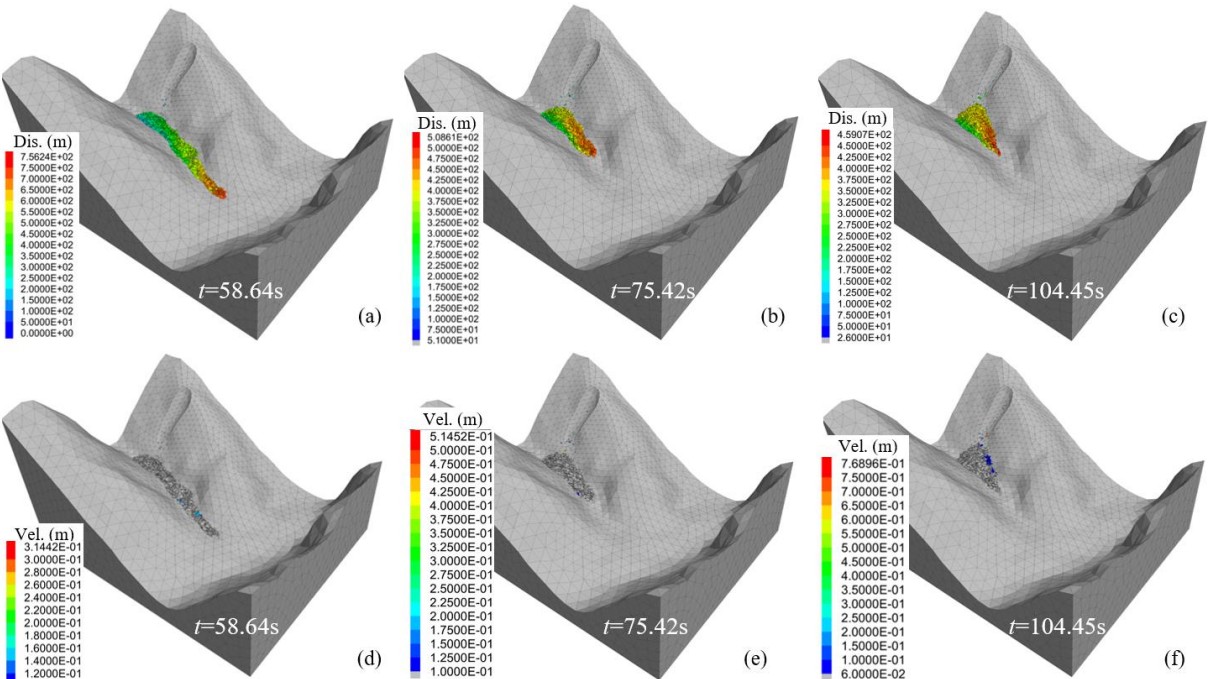


**Figure 11: The displacement and velocity of YXP rockslide with the internal friction angle of fractures of 8° (a, d), 12° (b, e) and 16° (c, f).**



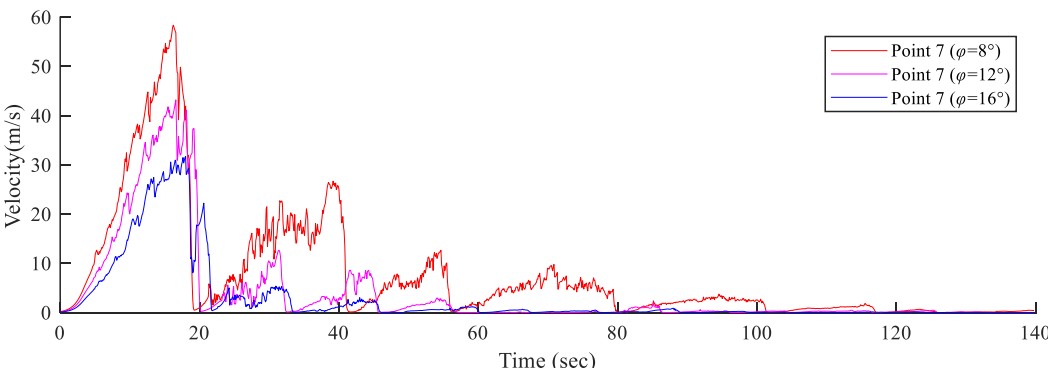

**Figure 12: The velocity monitoring results of point 9 under different internal fraction angle of fractures.**

## 4.2 Effect of fracture density

Fractures control the structure of rock mass, and the fractures density controls the size of rock cut by fractures. Physical experiments showed that different block sizes may have effects on the movement process and accumulation range of blocks (Lin et al. 2021; Lei et al. 2021). In order to study this effect, we set up four groups of fractures densities, which were 1/8, 1/4, 1/2 and 1 times of the original fracture density. For these models, the friction angle was set to 12°. The simulation results are shown in Fig.13. The larger the fracture density was, the more compact the deposit was, which was determined by the size of the blocks. With the increase of fractures density, the size of rock blocks decreased significantly. Due to the mutual restriction between large blocks, the movement time of the model with smaller fracture density was longer, and the duration was 158.26s, 141.17s, 125.33s and 104.45s, respectively (Fig.13). According to the previous analysis, when the fracture density was the original value, the rock blocks would still move downward along the gully with a slope of 14° (Fig. 9f). When the fractures density was 1/8, 1/4 and 1/2 of the original value, the deposit could stop in place, which showed that the size of the rock block of the deposit can affect the stability of the deposit, and the larger the rock block, the more stable the deposit. Therefore, the larger the fracture density, the faster the rock mass movement after failure. At the same time, the stability of the deposits is worse, which is easily taken by the debris flow.



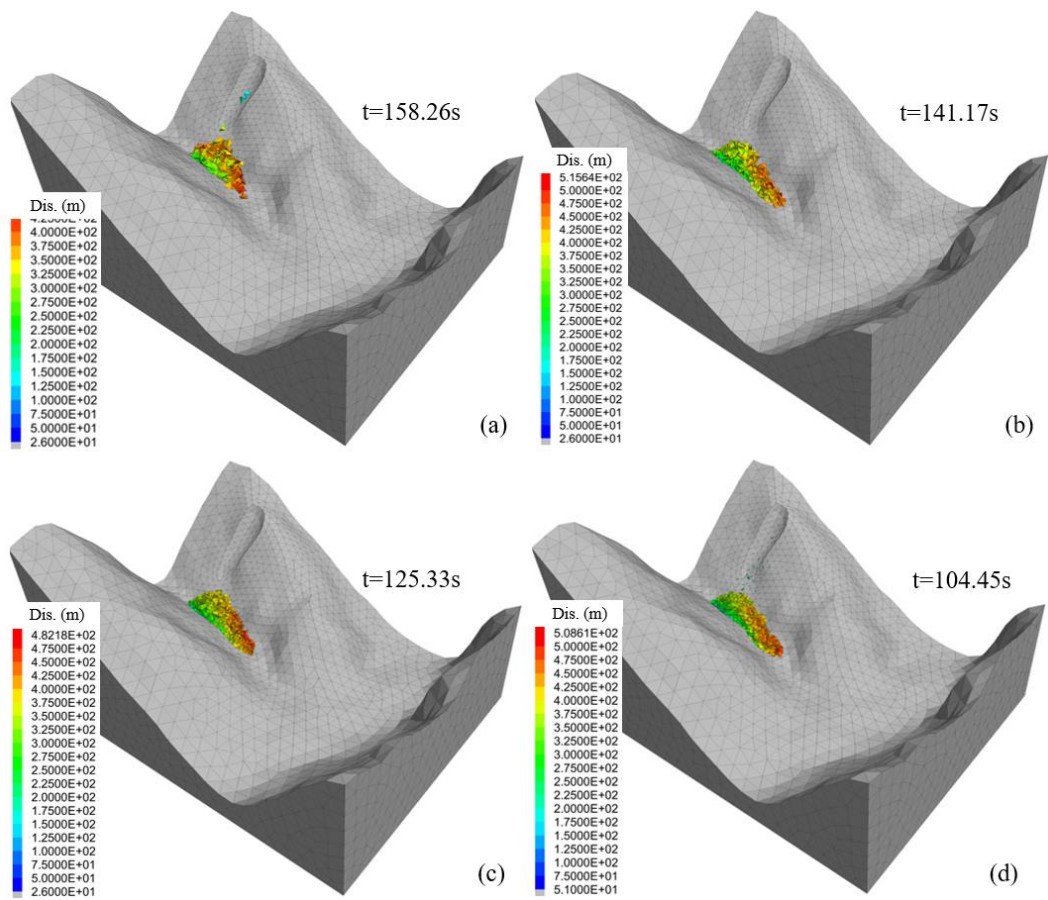

**Figure 13: The displacement and accumulation of YXP rockslide with fracture density of 1/8 (a), 1/4 (b), 1/2 (c) and 1 (d) times of the original value.**

## 4.3 The prevention work of check dam

The source area of post-earthquake rockslide is mostly located at the ridge of mountain. Therefore, the traditional prevention works such as setting lattice structure and bolt are not only expensive but also difficult to conduct. When the landslide materials

reach the bottom of the gully, the kinetic energy is obviously lost and easier to prevent these materials. Based on this guiding ideology, we set up a check dam at the downstream of the landslide deposit to prevent the further downward movement of the material. It can be seen from Fig. 8 that after setting the check dam, the landslide materials were obviously prevented in the gully. The field investigation also showed that check dam had good feasibility (Fig. 14c). However, it should be noted that if the check dam is full of rocks, the timely clean is essential. Otherwise, the whole gully will be blocked. In addition, some

outlets are needed in the check dam to drainage the water and small particles because there has perennial flow in the gully. The check dam proposed provided a novel perspective for prevention works of landslide on both sides of similar gully in meizoseismal area.



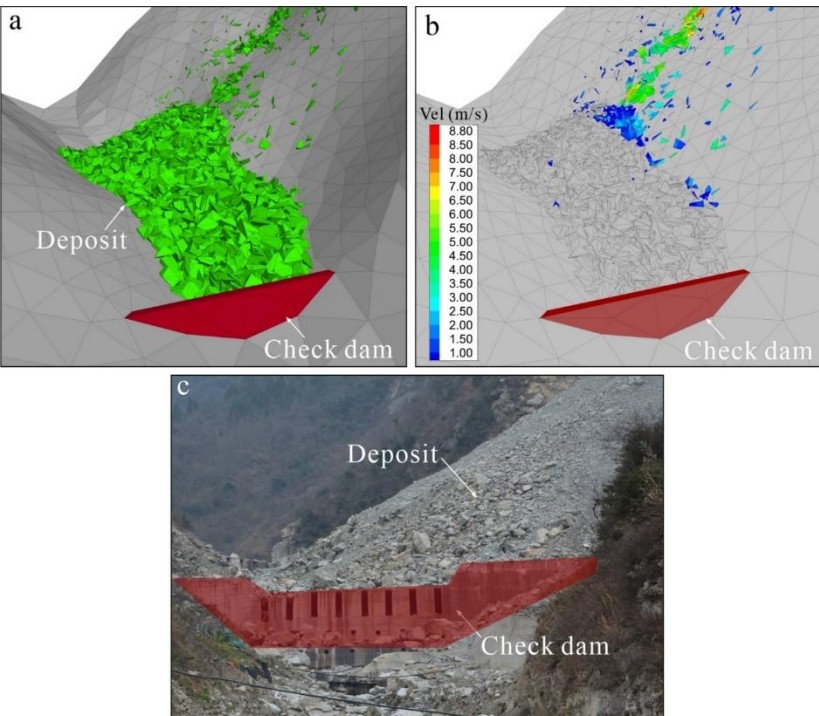

.

**Figure 14: The prevention works of check dam in downstream of depositional area. Pre- (a) and post-treatment (b) and the actual engineering (c).**

## 5 Conclusions

This paper presents a comprehensive study of the failure mechanism and dynamic characteristics of the YXP landslide. A field investigation and three-dimensional DEM were employed to analyse the effects of internal friction angle and fractures density on the simulation results. The main findings are summarized below.

(1) The YXP landslide failed and initiated under high pore-pressure in fractures caused by rainfall. The maximum velocity of the local block was 56.75 m/s, and the maximum displacement was and 508.61 m due to the steep terrain. Thus, the YXP landslide can be classified as a high-speed short-distance landslide.

(2) The blocks in the front edge of the sliding mass reached the bottom of the gully earlier, while the blocks in the rear edge deposits on the upper layer of the deposit later. Furthermore, if the bottom of the depositional area is inclined, the sliding mass may continue to move downward along the gully after reaching the gully bottom.

(3) The velocity of the sliding block and the depositional area increased correspondingly with the increase in the internal friction angle of the fractures. The size of the rock blocks can affect the stability of the deposit, and the larger the rock block, the more stable the deposit. Additionally, the larger the fracture density, the faster the velocity of rock mass after failure.



(4) The check dam is proposed as an appropriate prevention measure for the rockslide deposit in debris flow gully. By setting the check dam at the downstream of the deposit, the rock materials will be prevented from moving along the debris flow gully under the gravity.

## Author Contributions

Conceptualization, Y.Z and B.L.; methodology, J. L.; software, K.Y.; validation, K. H., J.C.; formal analysis, Z. Y.; investigation, J. L.; resources, X.Q.; data curation, J.C.; writing—original draft preparation, Y.Z.; writing—review and editing, J. C.; visualization, X.H.; supervision, X.H.; project administration, Y.Z.; funding acquisition, B.L., Y.Z. All authors have read and agreed to the published version of the manuscript.

## Declaration of competing interest

The authors declare that they have no known competing financial interests or personal relationships that could have appeared to influence the work reported in this paper.

## Acknowledgments

The authors would like to acknowledge the Science and Technology Research and Development Plan of China National Railway Group Corporation Limited (N2021G008), the China Postdoctoral Science Foundation (2024M753629), and the Fund Project of China Academy of Railway Sciences Group Corporation Limited (2023YJ190).

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
