# Peer review of "Kinetic characteristics investigation of the Yingxingping rockslide based on discrete element method combined with discrete fracture network"

_EGUsphere, 2024_

## Author Response (AR1)

**Authors' response**

Dear Editor,

RE: Manuscript number: EGUSPHERE-2024-2216

We would like to thank the journal Natural Hazards and Earth System Sciences for

giving us the opportunity to revise our manuscript.

We thank the reviewers for their careful read and thoughtful comments on previous

draft. We have carefully taken the comments into consideration in preparing our

revision, which has resulted in a paper that is clearer, more compelling, and broader.

The following summarizes how we responded to reviewers' comments.

Below is our response to the comments.

Thanks for all the help.

Best wishes,

All the authors.

**Response to Anonymous Referee 1#:**

The study investigates the development of rock fractures on mountain ridges in meizoseismal areas, which can lead to fatal rockslides based on site inquiries and aerial photography. A three-dimensional discrete element method (DEM) combined with a discrete fracture network (DFN) was employed to assess the dynamic process of the rockslide. The study analyzed the effects of fracture density and friction angle on the kinetic characteristics of the rockslide. This research provides valuable insights into the kinetic processes and prevention strategies for post-earthquake rockslides in meizoseismal areas. But several misunderstanding are encountered in this paper:

**1, In Section Introduction, please focus on scientific issues clearly and work to solve these problems well. This section needs definitely gives the research gap and objective about your work.**

**Response:** We have re-arranged the scientific problem of this paper, which is mainly how to combine discrete element method with discrete fracture network for accurate simulation of rockslides. At the same time, we have clarified the objectives of this work as follow: Therefore, the research objective of this paper is to use the discrete element method combined with discrete fracture network to achieve accurate inversion of rock landslides with fractures, reveal the instability mechanism and dynamic characteristics of a rockslide and the controlling effect of rock mass fracture parameters on rockslide, and explore the feasibility of using check dams to prevent such disasters. See section introduction.

- 2, Page 1 line 23 geological disasters or geo-hazards? please check all the text. Response: Thanks for your suggestions, we have unified the relevant vocabulary as geo-hazards throughout the text.
- 3, The phenomenon of cracks on the rear edge of the mountain before the landslide is common, whether it is rainfall, earthquakes, or gravity-induced landslide events, tends to form cracks at the trailing edge, but the formation of cracks is often a consequence rather than a cause. In other words, the formation of cracks is often the product of the beginning of the landslide, rather than the existing fracture and the landslide.

**Response:** Thank you for the reviewer's comments. Since the research object of this manuscript is located in a mountainous area with strong earthquakes, the mountains in this area are extremely shattered. After a landslide occurs, there are often still a large number of potentially unstable rock masses at the rear edge of the landslide, which still have the potential to slide. The YXP rockslide, the research object of this manuscript, is a typical example of such a rockslide. See section 2.1.

4, In Section Introduction, the author mentions several times that the type of fracture is important for landslide evolution, and the detection of fractures is also important, but the author does not mention this work in the manuscript.

Response: For the detection of rockfractures, we mainly use on-site measurement to

obtain more accurate fracture data. We collected more than 400 sets of fracture data in order to restore the true characteristics of rock fractures. See the second paragraph in section 3.2.

**5, There are many words in the text that the author has not explained, and it is impossible to understand and read. For example, page3 line 83, the author mentions three major faults? What is the meaning?**

**Response:** Thank you for the reviewer's suggestions. We have standardized the relevant descriptions in this article. The major thrust faults of Wenchuan earthquake include Wenchuan-Maoxian fault, Beichuan-Yingxiu fault and Anxian-Guanxian fault.

**6, Figure 1 shows the YXP landslide, while in the text uses the YXP rockslide? Why? Response: Thank you for the reviewer's reminder. According to the classification of landslide types by Hungr et al. (2014) (Hungr, O., Leroueil, S., & Picarelli, L. 2014. The Varnes classification of landslide types, an update. Landslides, 11(2), 167-194.), rockslide also belongs to landslide. For the sake of consistency, the corresponding positions in the whole text have been changed to rockslide. See figure 1 and the other**

**7, How was the data on the geohazards mentioned in Section 2.1 obtained? Or is it a reference to someone else's?**

part in the text.

**Response:** Thanks for the reviewer's comments. The location, geology, rainfall and other information in Section 2.1 were obtained through data surveys, while the evolution and scale of the disasters were obtained from on-site investigations after the geological disasters occurred.

**8, In chapter 2.2, the author directly gives the mechanism of the formation of the geohazard, which is only a simple description, without any data, how does the author make the mechanism clear?**

**Response:** Thank you for the reviewer's comments. In this section, we mainly aim to explain the cause of the rockslide through geological deduction methods. We have revised the title of this section to 'Formation mechanism'.

**9, The method mentioned by the author in Chapter 3 is the DEM method, which the author uses directly in Rockslide. And how is this data obtained? What is the reason for this value?**

**Response:** First, the important data of discrete elements method includes the terrain and the fracture parameters, and its core lies in the construction of fractures. We obtained 423 sets of fracture data through field investigation, and obtained two sets of main fractures of the data set through projection analysis. After that, the Monte Carlo random fracture network method was used, and a random fracture network was generated based on the two sets of main fractures. Then, the generated fracture network was used to cut the sliding body, so as to build a three-dimensional model close to the real rock mass characteristics. Finally, the discrete element method was used to define

the model and assign values for the block unit and fracture unit respectively, and the simulation study was carried out according to the required different working conditions. See section 3.

**10, There are two third parts of the text chapter? How to verify the reliability of the simulation, and how to use numerical simulation to reveal the gas process and prevent and control risks?**

Response: Thanks to the reviewer for reminding us, we have modified the title number in the text. Thanks for the reviewer's suggestion, the reliability of numerical simulation is the basis of the simulation in this paper. In order to ensure the reliability of the simulation, we first considered the real structural characteristics of the rock mass in the establishment of the model and constructed the rock mass fracture network based on the real fractures. Secondly, in terms of parameter selection, we selected the verified simulation parameters of the adjacent area (Liu et al. 2021). At the same time, the distance and range of the movement of the landslide obtained by numerical simulation are consistent with the actual situation, as shown in Figures 2 and 14, see section 3. The above three points can prove the reliability of the numerical simulation in this paper. In addition, if a prevention and control project is adopted, first of all, we can use numerical simulation to determine the layout location of the prevention and control project. Secondly, the velocity simulation results of the rockfall impact can provide a basis for the energy level of the prevention and control project, and the collapse range can provide a basis for the interception height of the prevention project. See section 5.3.

**Response to Anonymous Referee 2#:**

This manuscript describes field investigations and a numerical modeling study of a catastrophic post-earthquake rockslide in Wenchuan, China. On-site investigations and aerial imagery were utilized to characterize the landslide. A 3D discrete element method, integrated with a discrete fracture network, was employed to evaluate the dynamic processes of the rockslide. The research analyzed how fracture density and friction angle influenced the kinetic properties of the landslide.

Although this manuscript investigates an important landslide with advanced numerical modeling techniques, the scientific methods are not robust and the presented results and corresponding conclusions should be improved prior to publication. This paper provides some valuable insights into the parameterization of 3D discrete element models but should address several major issues as described below.

1. The main objectives of this manuscript should be clarified. The introduction begins by discussing the formation of ridge-top fractures during earthquakes as an important conditioning factor for subsequent slope stability. The authors don't provide any background information or substantiation that supports the studied landslide being related to the formation of cracks during the Wenchuan Earthquake, which occurred six years prior to the landslide failure. This should be discussed in section 2.1 – Geological Setting. In addition, the modeling investigations described in this paper do not simulate such processes (e.g., by simulating earthquake-induced damage), and it is not clear how this study contributes to a better understanding of earthquake-induced slope damage and the potential for subsequent landslides.

Response: Thank you very much for the reviewers' comments. The main purpose of this manuscript is to use the method of coupling discrete element and discrete fracture network to more accurately invert and reveal the instability mechanism and dynamic movement process of XGJ rockslide. Finally, the main factors affecting the post-destruction movement characteristics of the rockslide are discussed, including fracture strength, density and obstacles. In addition, the geological background mentioned by the reviewer, first of all, the area is located in the Longmenshan seismic belt on the eastern edge of the Qinghai-Tibet Plateau. At the same time, we added the historical image of YXP rockslide in 2009 after the Wenchuan Ms 8.0 earthquake as a comparison, see Figure 2a. The comparison between Figures 2a and 2b can prove that the damage area is constantly expanding. And there are many similar landslides formed by the Wenchuan earthquake in this area for reference, such as Daguangbao rockslide (Zhang et al. 2013), Xinmo Rockslide (Fan et al., 2017) and Sanxicun landslide (Pei et al. 2021). Therefore, we believe that the rear edge of the rockslide was generated by the earthquake, and the cracks continued to develop backwards after the earthquake.

2. Geologic Setting – The following points in this section should be clarified. Lines 83–88 please provide citations for descriptions of the earthquake, site geology, and regional climate. Please provide additional detail on how the total landslide area and thickness were measured. Is the reported thickness of 5.8 m in Line 96 measured from the thickness of depletion in the source area or thickness of

deposition? Please provide evidence or include references to substantiate the statement in Line 105 that "the landslide area was seriously broken" during the earthquake. What evidence do you have that these fractures did not exist prior to the earthquake? Please clarify the description of the failure plane in Line 112. Does the use of "clear" mean fully evacuated? What does "smooth" refer to?

**Response:** The total area of the rockslide was measured by using a drone aerial photography model. The thickness change of the sliding source area was obtained by differential analysis of the drone aerial photography elevation model and the ALOS satellite elevation. Finally, the average thickness of the sliding source area was obtained to be about 5.8m. The damage to the landslide area during the earthquake can be explained by the newly added historical images of the landslide area, see new Figure 2c. The description of the failure plane in Line 112 has been modified to "... the obvious sliding surface located at the rear edge of the YXP landslide was observed..."

3. Methods – The authors provide a detailed description of the equations used in the DEM formulation. However, there are several important aspects of the modeling methodology that are absent from the text. The authors choose to model bedrock and the moving landslide mass with different material properties without providing justification for this choice. In addition, the constitutive joint model is not specified, and the choice of material properties and joint strength properties are not substantiated. One reference is provided, but the assigned values are different than those used in this previous work. In addition, there is no reference or citation to the software package used when describing the equations used by the DEM software (e.g., Itasca Consulting Inc.), rather the authors cite their own prior work.

Response: Thank you for the reviewers' suggestions. This paper simplifies the sliding body and the sliding bed. Specifically, since the sliding bed remains stationary during movement and mainly acts on the contact surface with the sliding bed, we use elastic models for both the sliding body and the sliding bed in the modeling process. For the internal cracks of the rock mass, the Mohr-Coulomb joint model is used. Its characteristics are based on the Mohr-Coulomb strength criterion, which can judge the shear failure of the joint and is suitable for joint slip simulation in rock mass stability analysis. See paragraph 3 in section 3.2. Although we refer to the parameters of similar projects, different projects have their own unique characteristics, so we cannot use the same parameters completely, and some trial calculations are required. In terms of literature citations, we have supplemented the software manual of 3DEC 5.2 used in this manuscript and related references. See paragraph 2 in section 3.1.

4. Results—The authors report findings from a field investigation and conduct numerical modeling of the landslide. However, model results are not contextualized or compared with landslide conditions (e.g., deposit extent or thickness) and therefore it is unclear whether the numerical results are able to accurately reproduce this event.

Response: Thanks for the reviewer's suggestions. It is very important to compare the

simulation results with the rockslide, and we have taken this suggestion seriously. After analysis, we actually discuss this point in section 5.3. In section 5.3, we compare the simulation results after setting the dam with the actual results. The comparison results show that the numerical simulation is in good agreement with the actual situation in terms of accumulation characteristics, see Fig.14a, b. Since the YXP rockslide had a dam built in advance at the bottom when it failed, it is difficult to know what the accumulation range and maximum movement distance of the rockslide material would be without this dam. However, numerical simulation gives us this opportunity. By not setting the dam in advance in the numerical simulation, we can obtain the complete movement characteristics and final accumulation range of the sliding mass, which also illustrates the importance of setting the dam to prevent the sliding materials. Without the dam, the accumulation will continue to move down the channel for a long distance and accumulate in the channel, see Fig. 14c, d. Since the accumulation body is located in the debris flow channel, under heavy rainfall conditions, its stability may decrease rapidly and rush into the downstream villages along with the debris flow. In order to more intuitively compare the difference between the simulation results and the actual situation, we optimized Fig.14. See new Fig.14.

- 5. Discussion As written, this section presents further results which describe a parameter study on the impact of friction and angle and joint density on the model results. Differences in each model run are presented, but it is not clear which set of parameters best replicated the observed landslide features. While this is an interesting experiment, it is not meaningful in the context of this case study without further explanation and a comparison to the actual site conditions. It is not clear if parametrization for any of the other input material properties was conducted or how changes to these properties may impact model results. In addition, the presentation of results related to a check dam are not substantiated. **Response:** Thank the reviewers for their insightful suggestions. Indeed, in numerical simulation, many parameters may affect the simulation results, but these factors are divided into control factors and general factors. Through literature research and previous simulation test research, we believe that for rock landslides with developed fractures, the control factors are the parameters related to the fracture surface, among which the friction angle of the fracture surface and the degree of fracture development are particularly important. Therefore, these two factors are selected for comparative analysis, and the research on other factors needs further study. The discussion on the movement process of debris flow by retaining dams is based on the fact that conventional retaining measures are difficult to control such high-level rockslide disasters, so protection is chosen at the end of its movement path, and compared with the actual situation (Figure 14a, b).
- 6. The conclusions presented are valid observations from the modeling study but are not substantial in relation to the studied landslide or implications for similar future hazards. Conclusion #1 regarding the failure for the landslide under high pore-water pressure is not substantiated by the results of this study.

Response: We have revised the conclusion part of this paper as follow: (1) The YXP rockslide failed and initiated due to the decrease of rock strength under heavy rainfall. The maximum velocity of the local block was 56.75 m/s, and the maximum displacement was and 508.61 m due to the steep terrain. Thus, the YXP rockslide can be classified as a high-speed short-distance landslide. (2) By combining the discrete element method with discrete fracture networks, it is possible to accurately simulate and back-analyze the instability and movement processes of fractured rock landslides. In DEM simulation, the internal friction angle of fractures and fracture density significantly influence the movement velocity and deposit extent of rock landslides. A higher fracture density leads to faster instability of rock blocks, but it has no pronounced effect on travel distance. (3) The check dam is proposed as an appropriate prevention measure for the rockslide developed in debris flow gully. By setting the check dam at the downstream of the deposit, the rock materials will be prevented from moving along the debris flow gully under the gravity.

7. There are many places throughout the text where additional references are needed (see comment #2) or where the cited references are not relevant, for example see references in Line 40 to Evans et al., 2001 and Lines 52–54 to Mauka et al., 2017, or are missing from the reference list (e.g., Line 108 to Huang et al., 2014).

**Response:** Thanks to the reviewer for reminding us. We have carefully checked all references to ensure that there are no mis-citations or omissions.

**Response to Comment 1#:**

In this manuscript, the dynamic process of a rockslide was evaluated by threedimensional discrete element method (DEM) combined the discrete fracture network (DFN). This paper is well written. I think it can be accepted via minor revision. I have some comments which may be useful during the revision:

**1. The integration of field investigations and numerical simulations is a highly effective research approach. However, it is noted that the submission seems lacking comprehensive comparative analysis between the results obtained from field investigations and numerical simulations.**

**Response:** As the reviewer said, this paper adopts a method that combines field investigation with numerical simulation. By comparing the on-site drone aerial photography and previous satellite data, we can obtain the terrain changes and accumulation range before and after the YXP rockslide instability. The on-site fracture investigation can provide the distribution characteristic parameters of the fractures for numerical simulation, and the on-site boundary measurement can provide the numerical simulation of the instability scale and the final accumulation range and movement distance. Since we need to obtain the final potential movement range of the landslide, the first half of our simulation did not take into account the existing prevention and control projects. However, in Figure 14, we added a prevention and control project, and the simulation results are highly consistent with the site.

**2.Lines 140~183: it is not necessary to introduce the computing process of the DEM, instead, you may cite related papers.**

**Response:** We appreciate the reviewer's suggestion. We have deleted the DEM calculation process in the paper and added the corresponding literature, see section 3.1.

**3. The arrangement of Figure 6 should be rearranged to enhance the clarity of the numbers in Figure 6c.**

**Response:** Thank you for your suggestion. We have re-made Figure 6, optimized the layout of the whole figure, and placed Figure 6c alone in the lower left corner of Figure 6 to make it clearer. See Figure 6.

**Response to Comment #2:**

General comments:

In this submission, on-site inquiries and aerial photography were firstly utilized to study the basic characteristics of rockslides, and then the dynamic process of rockslide and the influences of fractures density and friction angle on kinetic characteristics were studied using the 3D discrete element method with the discrete fracture network. Generally speaking, this submission provides valuable information for assessing the dynamic process from on-site inquiries and numerical simulations, but there are still some issues that need to be revised to be published.

1. In the Introduction section, although there have been significant advancements in field investigations, theoretical models, and numerical simulations related to rockslides, the author should to highlight the innovative and necessary contributions of this submission's research.

**Response:** Thank you for your suggestion, and we have highlighted the innovative and necessary contributions of this study at the beginning and end of the Introduction.

2. Section 3.2, the bedrock and sliding rock mass were considered to be separate in the initial stage? Is the contact between the bedrock and sliding rock mass limited to frictional forces only? The authors should provide a detailed explanation.

Response: Thank you for the reviewer's question. This is a key question. I will explain it from the following aspects. First, in the discrete element simulation, all the rock masses are separated. However, since we mainly want to study the dynamic characteristics of the sliding body, we apply the "fix" command to the lower bedrock during the simulation. This command can ensure that the bedrock does not move in any direction, which is basically consistent with reality. Secondly, since the bedrock and the sliding body have different mechanical properties, and the shear strength of the sliding body is relatively lower, there is a strength difference between the two materials. At the same time, there is also a structural surface between the sliding body and the sliding bed. Therefore, the parameters of the structural surface at the contact surface between the two directly control the sliding of the upper rock mass. In the simulation, the parameters of the structural surface include Cohesion, Friction angle, Normal stiffness, Shear stiffness, etc., which jointly control the stability of the sliding body. Therefore, it is not just the friction parameters that control the sliding of the sliding body.

**Response to Comment 3**

General comments

Very good research on rock mechanics and discrete fracture network. Please, follow my specific comments to improve the manuscript.

**Specific comments**

- 1. Lines 64-66. "Discrete Fracture Network (DFN) models are primarily based on the establishment of joint and fracture networks... cannot be fully measured". Insert recent review papers on the link between field surveys and modelling aspects of Discrete Fracture Network models:
- Medici G, Ling F, Shang J 2023. Review of discrete fracture network characterization for geothermal energy extraction. Frontiers in Earth Science 11, 1328397.
- Kolapo P, Ogunsola NO, Munemo P, Alewi D, Komolafe K, Giwa-Bioku A 2023. DFN: an emerging tool for stoc hastic modelling and geomechanical design. Eng 4(1), 174-205.

**Response:** Thank you for your suggestion. We have added the relevant literature on DFN in the article. See section introduction.

**2. Line 79. Specify the 3 to 4 specific objectives of your research by using numbers.**

**Response:** We sincerely appreciate the reviewer's constructive suggestions. In response, we have reorganized the research objective of this paper as follow: The research objective of this paper is to use the discrete element method combined with discrete fracture network to achieve accurate inversion of rock landslides with fractures, reveal the instability mechanism and dynamic characteristics of a rockslide and the controlling effect of rock mass fracture parameters on rockslide, and explore the feasibility of using check dams to prevent such disasters. See section introduction.

**3. Line 83. "Three major faults". Please, specify the type of faults. Normal, strike-slip?**

**Response:** Thanks, it should be thrust fault.

**4. Lines 321-334. The bulletin points are 3 so the specific objectives should be 3 to match.**

**Response:** We have revised the conclusions and aligned them with the specific objectives. See section conclusions.

**5. Lines 321-334. Add a "take home" message for the researchers working in your field.**

**Response:** We have reorganized the conclusion section to make it more concise.

**6. Lines 350-442. Please, integrate relevant literature on DFN.**

**Response:** Based on the revised introduction, we have modified the corresponding references.

**7. Figures and tables**

Figure 1. Make the figure larger.

Figure 6c. The stereonet that shows the two sets of fractures should be much larger.

Figure 6c. Consider to make it a separate figure.

Figure 14. Fractures are very difficult to see. Increase the graphic resolution.

Figure 14. Make also the figure larger, there is room for this change.

**Response:** According to the reviewer's suggestion, we have enlarged Figure 1 and Figure 6C separately. At the same time, we have revised and optimized Figure 14 and improved the resolution of Figure 14.